# Evaluation of a Porcine Endogenous Reference Gene (Internal Sample Control) in a Porcine Reproductive and Respiratory Syndrome Virus RT-qPCR

**DOI:** 10.3390/vetsci10060381

**Published:** 2023-05-31

**Authors:** Berenice Munguía-Ramírez, Betsy Armenta-Leyva, Alexandra Henao-Díaz, Fangshu Ye, David H. Baum, Luis G. Giménez-Lirola, Jeffrey J. Zimmerman

**Affiliations:** 1Veterinary Diagnostic and Production Animal Medicine Department, College of Veterinary Medicine, Iowa State University, Ames, IA 50011, USA; betsyarl@iastate.edu (B.A.-L.); dhbaum@iastate.edu (D.H.B.); luisggl@iastate.edu (L.G.G.-L.); jjzimm@iastate.edu (J.J.Z.); 2Pig Improvement Company (PIC) México, Santiago de Querétaro 76040, Querétaro, Mexico; alexandra.henao@genusplc.com; 3Department of Statistics, College of Liberal Arts and Sciences, Iowa State University, Ames, IA 50011, USA; fye@iastate.edu

**Keywords:** endogenous reference genes, internal sample control, PRRSV, RT-qPCR, diagnostics

## Abstract

**Simple Summary:**

Endogenous reference genes in diagnostic specimens are used to monitor sample quality in a quantitative polymerase chain reaction (qPCR), i.e., serve the role of internal sample controls (ISC). However, there is little information on the consistency of ISC expression among specimen types or the interpretation of ISC results. Therefore, the aim of this study was to evaluate the expression of a porcine-specific ISC in serum, oral fluid, and fecal specimens collected from pigs of known porcine reproductive and respiratory syndrome virus (PRRSV) infection status and tested using a commercial PRRSV reverse transcription-qPCR. The ISC was detected in 100% of the specimens tested and was not affected by PRRSV infection status of the pigs, but ISC concentration varied between specimen types. Thus, reference limits were established for each specimen to provide guidelines for ISC interpretation. Overall, the ISC evaluated herein can be used to accurately monitor sample quality in swine specimens tested for PRRSV. In particular, failure to detect the ISC indicates an irregularity with the sample or the testing procedures.

**Abstract:**

Endogenous reference genes are used in gene-expression studies to “normalize” the results and, increasingly, as internal sample controls (ISC) in diagnostic quantitative polymerase chain reaction (qPCR). Three studies were conducted to evaluate the performance of a porcine-specific ISC in a commercial porcine reproductive and respiratory syndrome virus (PRRSV) reverse transcription-qPCR. Study 1 evaluated the species specificity of the ISC by testing serum from seven non-porcine domestic species (*n* = 34). In Study 2, the constancy of ISC detection over time (≥42 days) was assessed in oral fluid (*n* = 130), serum (*n* = 215), and feces (*n* = 132) collected from individual pigs of known PRRSV status. In Study 3, serum (*n* = 150), oral fluid (*n* = 150), and fecal samples (*n* = 75 feces, 75 fecal swabs) from commercial herds were used to establish ISC reference limits. Study 1 showed that the ISC was porcine-specific, i.e., all samples from non-porcine species were ISC negative (*n* = 34). In Study 2, the ISC was detected in all oral fluid, serum, and fecal samples, but differed in concentration between specimens (*p* < 0.05; mixed-effects regression model). The results of Study 3 were used to establish ISC reference limits for the 5th, 2.5th and 1.25th percentiles. Overall, the ISC response was consistent to the point that failure in detection is sufficient justification for re-testing and/or re-sampling.

## 1. Introduction

In basic gene expression research, quantitative polymerase chain reaction (qPCR) cycle threshold (Cq) values of the gene(s) of interest are commonly expressed relative to the response of one or more endogenous reference genes, i.e., housekeeping genes inherent to the host-derived specimen [1,2,3,4]. 

In diagnostic qPCR, endogenous reference genes may be used as internal sample controls (ISC) that are amplified and detected together with the target of interest. Because they are subjected to the same conditions as the diagnostic target, consistent detection of the ISC would provide assurance that the overall process, i.e., sample collection through testing, was performed correctly [5,6,7]. Housekeeping genes frequently used as ISCs include glyceraldehyde 3-phosphate dehydrogenase, β-actin, 18S rRNA, peptidyl-prolyl cis-trans isomerase, β-2-microglobulin, and ubiquitin C [8,9]. Although not widely investigated, the use of ISCs in veterinary diagnostic has been reported for several species, e.g., 18S rRNA for the DNA detection of *Salmonella enterica* in cattle lymph nodes [10], bird β-actin for the RT-qPCR detection of avian influenza virus [6], and porcine β-actin for the detection of African swine fever virus by qPCR [11]. 

Ideally, ISCs should be constantly expressed across specimens and regardless of the physiological conditions of the individual. Nonetheless, they rarely meet these criteria [12,13]. The underlying assumption that ISCs are consistently present in all specimens provides a benchmark against which to measure changes in the target of interest [14]. Thus, it must be established that ISCs meet the requirement for constancy across specimens if they are to be used for quality control purposes in diagnostic qPCR testing. Herein, a porcine-specific ISC RNA included in a commercial porcine reproductive and respiratory syndrome virus (PRRSV) reverse transcription (RT)-qPCR (IDEXX Laboratories, Inc., Westbrook, ME, USA) was evaluated for its expression in swine serum, oral fluid, and fecal specimens. The gene targeted by this ISC is proprietary information, but the product insert describes the target as an “endogenous host RNA” that is amplified simultaneously with PRRSV.

## 2. Materials and Methods

### 2.1. Experimental Design

Three studies were performed. Study 1 tested samples from non-porcine species to assess the porcine specificity of the ISC. Specifically, five samples each from seven domestic non-porcine species (*n =* 34) were tested for the presence of the ISC to assess its exclusivity for swine. Study 2 used samples collected from animals under experimental conditions (all procedures approved by the Iowa State University Office of Research Ethics, Institutional Animal Care and Use Committee, IACUC) to evaluate the constancy of ISC expression over time. That is, oral fluid (*n =* 130), serum (*n =* 215), and feces (*n =* 132) collected from individual pigs of known PRRSV status for a minimum of 42 days were tested to evaluate the expression of the ISC RNA over time. Study 3 used specimens submitted for routine diagnostic testing to establish the expected reference intervals in serum, oral fluid, and fecal samples. Specimens included oral fluid (*n =* 150), serum (*n =* 150), and fecal samples (*n =* 75 feces, 75 fecal swabs) submitted to the Iowa State University Veterinary Diagnostic Laboratory (ISU-VDL) for routine diagnostic testing. A commercial PRRSV RT-qPCR (RealPCR^TM^NA PRRS Types1-2 RNA Mix, IDEXX Laboratories, Inc., Westbrook, ME, USA) containing primers and probes for the porcine ISC was used throughout. 

For data analysis, Cqs were converted to “efficiency standardized Cqs (ECqs)” [15,16], and then transformed to the cube root. Notably, and in direct contrast to Cqs, ECq values increase as the concentration of the target in the sample increases. Data were analyzed using a mixed-effects regression model (Study 2) and a linear regression model (Study 3) (R v.4.2.1, https://www.r-project.org/ (accessed on 5 September 2022)). In addition, reference intervals for ISC responses in serum, oral fluid, and fecal samples were calculated from Study 3 results, following the recommendations of the American Society for Veterinary Clinical Pathology [17]. 

### 2.2. Study 1—ISC Porcine Specificity

The porcine specificity of the ISC was evaluated by testing 34 serum samples from seven non-porcine domestic species, i.e., 4 or 5 samples each from avian, bovine, equine, caprine, ovine, canine, feline samples. Non-porcine serum samples were selected from specimens submitted to the ISU-VDL for routine diagnostic testing. No criteria were used to select samples aside from the requirement for a volume sufficient for testing. In addition, one porcine serum sample was included as a known positive control. Samples were randomly ordered prior to testing. 

### 2.3. Study 2—ISC over Time in Individual Pigs

The expression of the ISC over time in individual pigs was evaluated in specimens collected from PRRSV viremic and non-viremic pigs under experimental conditions. Sample sets originated from two experiments (Set 1 and Set 2). Within sets and specimens, samples were randomized prior to testing. 

Samples in Set 1 included oral fluid (*n =* 130), serum (*n =* 132), and fecal samples (*n =* 132) from a study conducted with the approval of the Iowa State University Office of Research Ethics, Institutional Animal Care and Use Committee (IACUC, Log # 3-16-8214-S) involving 12 individually housed 14-week-old pigs vaccinated with a PRRSV modified-live virus vaccine (Ingelvac^®^ PRRS MLV, Boehringer Ingelheim Vetmedica, Inc., Duluth, GA, USA). As fully described elsewhere [18], 14-week-old PRRSV-naive pigs (*n =* 12) housed in a biosafety level 2 (BSL-2) livestock infectious disease isolation facility (Iowa State University, Ames, Iowa, USA) accredited by the Association for Assessment and Accreditation of Laboratory Animal Care were intramuscularly vaccinated with a modified-live virus PRRSV vaccine (2 mL, Ingelvac^®^ PRRS MLV, Boehringer Ingelheim Vetmedica, Inc., Duluth, GA, USA). Set 1 date- and pig-matched blood, oral fluid, and fecal samples were collected on days post-vaccination (DPVs) −7, 0, 3, 6, 9, 14, 17, 21, 28, 35, 42, processed, and stored in aliquots at −80 °C. 

Set 2 consisted of serum samples (*n =* 84) collected from a study performed with the approval of the Iowa State University Office of Research Ethics, IACUC (Log #9-04-5751-S), comprising 10 three-week-old PRRSV-naive pigs housed in a BSL-2 Livestock Infectious Disease Isolation facility (Iowa State University, Ames, IA, USA). As fully described elsewhere [19], pigs were intramuscularly inoculated with 1 mL (1 × 10^4^ TCID_50_/mL) of PRRSV isolate ATCC VR-2332 (American Type Culture Collection, Manassas, VA, USA). Serum samples used for this study were from seven pigs at days post inoculation (DPI) −6, 7, 42, 56, 70, 98, 112, 126, 140, 154, 168 and 182. 

### 2.4. Study 3—ISC in Field Samples

Study 3 samples consisted of serum (*n =* 150; 75 PRRSV RNA positive), oral fluids (*n =* 150; 75 PRRSV RNA positive), and fecal samples (*n =* 150; unknown PRRSV status) submitted to the ISU-VDL from commercial swine herds for routine PRRSV RT-qPCR testing (serum and oral fluids) or enteric disease testing, i.e., porcine epidemic diarrhea virus, transmissible gastroenteritis virus, and/or porcine deltacoronavirus (fecal specimens). Both feces (*n =* 75) and fecal swabs (*n =* 75) were included in Study 3 because both sample types are routinely received for diagnostic testing. After receipt at the ISU-VDL, fecal specimens were processed by combining ~2 g of sample with 2 mL of PBS (Gibco^TM^, Thermo Fisher Scientific, Waltham, MA, USA) in a 5 mL tube (Falcon^TM^, Fisher Scientific, Pittsburgh, PA, USA) and vortexing for ~5 s. Fecal swabs were processed by placing swabs and 2 mL of PBS (Gibco^TM^, Thermo Fisher Scientific, Waltham, MA, USA) in a 5 mL tube (Falcon^TM^, Fisher Scientific, Pittsburgh, PA, USA) and vortexing for ~5 s. As is common for samples received by diagnostic laboratories, the conditions under which samples were collected, the status of the farms of origin, and the handling procedures post-collection were unknown. Samples were randomly ordered within specimens prior to testing.

### 2.5. Nucleic Acid Extraction and Amplification

All sample handling and laboratory procedures were performed in a certified biological safety cabinet using calibrated (in-date) pipettes. Extraction controls (positive and negative) were included in each run and amplification controls (positive and negative) were included on each qPCR plate (RealPCR^TM^ Positive Control, IDEXX Laboratories, Inc., Westbrook, ME, USA). In addition, specimen-specific (serum, oral fluid, and feces) and target specific (PRRSV and ISC) reference standards (*n =* 4) were included on each plate. Reference standards were created by rehydrating a lyophilized vial of PRRSV MLV (10-dose, Ingelvac^®^ PRRS MLV) with 20 mL of a PRRSV-negative specimen as diluent (serum, oral fluid, or a fecal suspension). In this case, the fecal suspension (20% *w*/*v*) was generated by resuspending 10 g of feces with 50 mL of PBS (Gibco^TM^, Thermo Fisher Scientific, Waltham, MA, USA) in a 50 mL tube (Nunc^TM^, Thermo Fisher Scientific, Rochester, NY, USA). The suspension was vortexed for 20 s, centrifuged at 3300× *g* for 3.5 h, and the supernatant used to resuspend the vaccine. The PRRSV MLV resuspended with the corresponding PRRSV-negative diluent was subjected to tenfold dilutions, and the 1 × 10^4^ dilution was used as a reference standard.

Total RNA extraction from serum, oral fluid, and fecal specimens was performed using the RealPCRTM DNA/RNA Spin Column Kit (IDEXX Laboratories, Inc., Westbrook, ME, USA) as directed by the manufacturer. For serum, 5 µL of proteinase K was added to each 2 mL tube (one per sample) containing 200 µL of the sample. The mixture was vortexed for 10 s and centrifuged at 8000× *g* for 30 s. Lysis working solution was prepared by mixing 195 µL of lysis buffer with 5 µL of carrier RNA per sample. For oral fluids and fecal specimens, lysis working solution was prepared using 190 µL of lysis buffer, 5 µL of carrier RNA, and 5 µL of proteinase K for each sample to be extracted. Thereafter, 200 µL of the respective specimen, i.e., oral fluids, fecal swabs, or the supernatant of fecal samples previously centrifuged at 3000× *g* for 5 min, were added.

The following procedures were identical for all three specimens. Samples were mixed with 200 µL of the lysis working solution, incubated for 3 min at 25 °C, and centrifuged (8000× *g*, 30 s). Ethanol (200 µL) was added to the mixture, vortexed for 10 s, incubated (5 min at 25 °C), and then centrifuged (8000× *g*, 30 s) to settle the contents. Thereafter, the mixture was transferred to a spin column containing a silica membrane to bind nucleic acids. Columns were centrifuged (8000× *g*, 3 min), then washed with wash solution 1 (400 µL) and wash solution 2 (400 µL) with centrifugation at 11,000× *g* for 30 s between each wash. Columns were subjected to a final wash with wash 2 (200 µL) and centrifuged (20,000× *g*) for 2 min. To elute nucleic acid from the column, elution water (50 µL at 70 °C) was added to the column followed by a short incubation (1 min at 25 °C), and centrifugation (20,000× *g*, 1 min). The eluate was collected in a 2 mL tube and immediately subjected to RT-qPCR. 

The PRRSV RT-qPCR reaction (25 µL) contained 10 µL of RNA Master Mix (RealPCR^TM^ RNA Master Mix, IDEXX Laboratories, Inc., Westbrook, ME, USA), 10 µL of PRRSV RNA target mix (RealPCR^TM^ PRRS Types 1-2 RNA target Mix, IDEXX Laboratories, Inc., Westbrook, ME, USA) which also contained the ISC primers and probe, and 5 µL of the extracted nucleic acids. The RT-qPCR run was performed using a Magnetic Induction Cycler qPCR (Mic qPCR Cycler, Bio Molecular Systems, Queensland, Australia) and read using the Mic qPCR Cycler Software v.2.10.4 (Bio Molecular Systems, Queensland, Australia). The cycling program consisted of 45 cycles of reverse transcription at 50 °C for 15 min, denaturation at 95 °C for 15 s, and two amplification steps (95 °C for 15 s, 60 °C for 30 s). 

### 2.6. Data Analysis

For data analysis, RT-qPCR Cq results were re-expressed as “efficiency standardized Cqs (ECqs)”, as shown in Equation (1) [15,16]. It should be borne in mind that, unlike Cqs, ECq values increase as the concentration of the target in the sample increases.
ECq = E^−ΔCq^ = E^−(Cq Sample—Mean Cq of reference standards)^(1)

In Equation (1), “E” indicates the mean amplification efficiency of the four reference standards expressed as a ratio from 1 to 2, and ΔCq represents the difference between the Cq of the sample and the mean Cq of the four reference standards on each plate. Amplification efficiency can be expressed as either a percentage or as the number of amplicons at the end of each cycle divided by the number of amplicons at the beginning, i.e., a ratio between 1 and 2, where 2 indicates 100% amplification efficiency. As amplification efficiency calculated by the Mic qPCR Cycler Software v.2.10.4 (Equation (2)) was reported as a percentage (values between 0 and 1), a value of 1 was added to the resulting mean amplification efficiency of the four reference standards in order to express it as a ratio from 1 to 2 and calculate ECqs in Equation (1). Analysis of the ECq data was done using R v.4.2.1 (https://www.r-project.org (accessed on 5 September 2022)).
Efficiency (E) = 10^−1/Slope^ − 1(2)

For Study 2, the effect of time on the expression of the ISC was evaluated for each specimen (serum, oral fluid, feces) using a mixed-effects regression model with “DPV” and “specimen” as fixed effects in the model for Set 1, and “DPI” as a fixed effect for Set 2. The effect of time was measured as the estimated change in ISC ECq per “DPV” (Set 1) or “DPI” (Set 2) as described by the slope of the regression lines. 

For Study 3, a linear regression model with “specimen” as the fixed effect was fitted. The overall difference in ISC ECqs between serum, oral fluid, and fecal specimens was assessed using a Type-III analysis of variance (ANOVA) from the fitted model. To evaluate the effect of PRRSV status on the response of the ISC, a linear regression model with “PRRSV ECq” as the explanatory variable was fitted for oral fluid and serum data. Field sample test results were used to establish guidelines for the interpretation of ISC ECq values by calculating reference intervals for each specimen type (serum, oral fluid, feces, and fecal swabs) using “stats” and “boot” packages in R v.4.2.1 (https://www.r-project.org/ (accessed on 5 September 2022)). In particular, values at or below the lower reference bounds denote unusually low ISC concentration and suggest the need for re-testing and/or re-sampling. Following the recommendations of the American Society for Veterinary Clinical Pathology for sample size > 120 [17], serum and oral fluid reference limits for the 5th, 2.5th, and 1.25th percentiles and their 95% confidence intervals (CI) were calculated using a non-parametric percentile method, i.e., quantile estimation. For sample sizes < 120, i.e., feces *n =* 75 and fecal swabs *n =* 75, reference limits were calculated using the non-parametric method and their 95% CIs were calculated using the bootstrap method.

## 3. Results

Overall, 158/282 (56.0%) sera, 146/280 (52.1%) oral fluids, and 27/282 (9.6%) fecal samples were positive for PRRSV RNA. Although there are only a few studies describing PRRSV shedding in fecal samples, all reports agree on intermittent detection and/or isolation of PRRSV in this specimen [20,21,22,23]. 

### 3.1. Study 1—ISC Porcine Specificity

All serum specimens from non-porcine species were negative for the ISC (*n =* 34), with only the positive control showing an amplification signal. Thus, based on the data, the PRRSV RT-qPCR ISC was porcine-specific. 

### 3.2. Study 2—ISC over Time in Individual Pigs

All pigs from sample Set 1 (*n =* 12) were negative for PRRSV at −7 and 0 DPV, with the first positive result at 3 DPV (Figure 1A). Frequency of PRRSV detection in pigs was higher in serum and oral fluids compared to feces (Figure 1B). 

The ISC was detected in all samples at all DPVs (*n =* 394). ISC ECqs were not affected by DPVs (*p* > 0.05; mixed-effects regression model) (Table 1), but the analysis detected a difference in ISC ECqs between specimens (*p* < 0.05; Type-III ANOVA). In sample Set 2, all pigs (*n =* 7) were positive for PRRSV RNA on DPI 7 and negative on DPIs −6, 42, 56, 70, 98, 112, 126, 140, 154, 168 and 182. The mixed-effects regression model detected no “DPI effect” on serum ISC ECqs (*p* > 0.05) and estimated the slope of the line at 0.00 ECqs per day (95% CI: −0.00, 0.00).

### 3.3. Study 3—ISC in Field Samples

The ISC was detected in all field specimens (*n =* 928) with the exception of two PRRSV-negative samples (one serum, and one oral fluid). These two samples were ISC positive upon re-testing. ISC ECq values differed between serum, oral fluid, and fecal field specimens (*p* < 0.05; Type-III ANOVA). By specimen, the mean ISC ECqs (95% CI) were 2.33 (2.17, 2.49) for serum, 0.83 (0.67, 0.99) for oral fluid, 1.87 (1.64, 2.10) for feces, and 0.97 (0.74, 1.20) for fecal swabs. The concurrent detection of PRRSV RNA had no effect on ISC ECqs in serum and oral fluid samples (*p* > 0.05; linear regression model). In fecal specimens, PRRSV was detected in five samples (four feces, one fecal swab), with ECqs of 1.11, 0.59, 0.39, 0.23, and 0.32, respectively. ISC ECq reference limits (95% CIs) for the 5th, 2.5th, and 1.25th percentiles are given in Table 2 for serum, oral fluid, feces, and fecal swabs.

## 4. Discussion

Quality controls used in routine diagnostic RT-qPCR testing to monitor irregularities in the process and assure consistent results include positive/negative extraction controls and positive/negative amplification controls. However, these controls account only for testing anomalies, e.g., contamination, between-technician variation, pipetting errors, issues with extraction and/or amplification reagents, problems with the equipment, etc., and do not account for issues associated with the sample itself. Housekeeping genes offer the potential to address this shortcoming.

Housekeeping genes have been used as ISCs in human [24,25,26] and veterinary diagnostic research [6,10,11] because they are essential to basic cellular functions and are, therefore, innate to the biological specimens being tested [27]. For that reason, failure to detect the ISC would indicate a fault at some point between sample collection and final PCR testing. However, care needs to be taken in selecting ISCs because the expression of housekeeping genes varies among cell types, as a function of disease status, and by age and/or gender [28,29,30,31]. For example, viral infections and resultant cellular changes can affect overall gene expression [32]. Furthermore, a study involving influenza A virus infection demonstrated that 18S rRNA was more stable than other housekeeping genes included in the evaluation, e.g., ACTB, GADPH, ATP synthase, H+ transporting, mitochondrial F1 complex, beta polypeptide (ATP5B), and ATP synthase, H+ transporting, mitochondrial Fo complex, subunit C1 (subunit 9) (ATP5G1), in human bronchial epithelial cells, pig tracheal epithelial cells, and avian lung cells [12]. 

Because there is no single best ISC housekeeping gene, the constancy of candidate housekeeping gene expression must be carefully evaluated before being used as a qPCR control [33,34]. In this study, the porcine-specific RNA ISC was consistently detected over time in serum, oral fluid, and fecal specimens from pigs vaccinated with MLV PRRSV or inoculated with wild-type PRRSV under experimental conditions and in all field specimens, i.e., serum, oral fluid, fecal samples, and fecal swabs, regardless of PRRSV status. Mean ISC ECqs did not differ between serum samples collected under experimental conditions versus in the field, but the mean ISC ECqs for oral fluid and fecal specimens were lower in field samples than in samples collected under experimental conditions. This may be because research samples were collected in a controlled environment and quickly processed for storage, whereas the sampling history of field samples was unknown. Previous data have shown decay in ISC RNA (IDEXX Laboratories, Inc., Westbrook, ME, USA) in oral fluids subjected to ≥2 freeze-thaw cycles, and in oral fluid and fecal samples exposed to ≥4 °C for more than 24 h [35].

## 5. Conclusions

This study evaluated the detection of a porcine-specific ISC in a commercial PRRSV RT-qPCR and provided guidelines for the interpretation of ISC results by calculating reference limits for serum, oral fluid, feces, and fecal swabs collected from pigs under experimental and field conditions. The ISC was not detected in avian, bovine, equine, caprine, ovine, canine, or feline serum samples; i.e., among domestic animal species, it appears to be exclusive to swine. In swine specimens collected from the same individuals, the ISC was consistently detected over time in all specimens and was not affected by PRRSV infection. Overall, among 928 porcine samples tested (366 serum, 280 oral fluids, 282 fecal samples), the ISC was detected in 926, but the 2 ISC negative samples (one serum and one oral fluid) were positive upon re-testing. Thus, failure to detect the ISC in swine specimens suggests a significant irregularity with the sample or the testing procedures and, in either case, re-testing and/or re-sampling should be carried out. 

## Figures and Tables

**Figure 1 vetsci-10-00381-f001:**
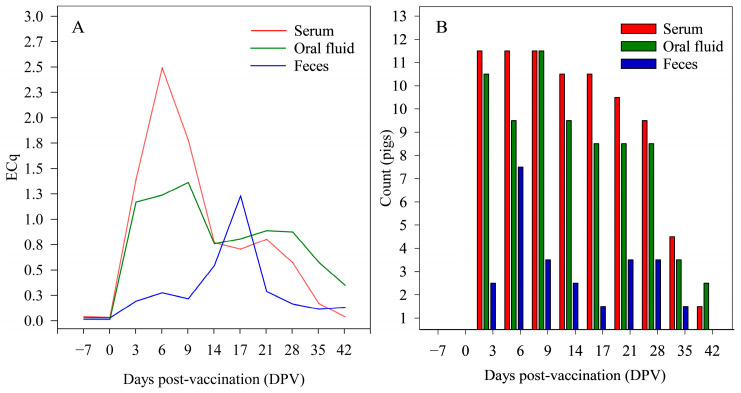
(**A**) Mean porcine reproductive and respiratory syndrome virus (PRRSV) efficiency standardized Cqs (ECq) values and (**B**) frequency of PRRSV detection (PRRSV and ISC RT-qPCR; IDEXX Laboratories, Inc.) by day post-vaccination (DPV) in serum (*n =* 132), oral fluid (*n =* 130), and fecal samples (*n =* 132) from pigs (*n =* 12) vaccinated with a PRRSV modified-live vaccine (Ingelvac^®^ PRRS MLV) (Study 2, Set 1).

**Table 1 vetsci-10-00381-t001:** Effect of time on internal sample control (ISC) efficiency standardized Cqs (ECqs) estimated by a mixed-effects regression model (R v.4.2.1, https://www.r-project.org/ (accessed on 5 September 2022)) in oral fluid (*n =* 130), serum (*n =* 132), and fecal samples (*n =* 132) collected from individual pigs (*n =* 12) over 49 days (Study 2, Set 1) and tested with a PRRSV and ISC RT-qPCR (IDEXX Laboratories, Inc., Westbrook, ME, USA).

Specimen	Intercept ^1^	Slope ^2^ (95% CI)	*p*-Value
Serum	1.80	0.00 (−0.00, 0.00)	>0.05
Oral fluid	2.13	0.00 (0.00, 0.01)	>0.05
Feces	1.31	−0.00 (−0.00, 0.00)	>0.05

^1^ Initial ISC ECq. ^2^ ISC ΔECq by day post-vaccination.

**Table 2 vetsci-10-00381-t002:** Efficiency standardized Cq (ECq) internal sample control (ISC) reference limits for serum (*n =* 150), oral fluids (*n =* 150), and fecal samples (*n =* 150) collected from the field (Study 3) and tested with a PRRSV and ISC RT-qPCR (IDEXX Laboratories, Inc., Westbrook, ME, USA).

	ECqs (95% CI) ^1^		
Specimen	5th Percentile	2.5th Percentile	1.25th Percentile
Serum	1.26 (1.12, 1.40)	1.16 (1.10, 1.27)	1.11 (0.56, 1.23)
Oral fluid	0.28 (0.22, 0.44)	0.25 (0.09, 0.36)	0.11 (0.03, 0.26)
Feces ^2^	0.49 (0.18, 0.60)	0.37 (0.17, 0.55)	0.18 (0.17, 0.52)
Fecal swab ^2^	0.27 (0.04, 0.44)	0.14 (0.03, 0.39)	0.04 (0.03, 0.34)

^1^ ECq reference limits and their 95% confidence intervals (CI) were calculated using the non-parametric percentile method (R v.4.2.1, https://www.r-project.org/ (accessed on 5 September 2022)), following the recommendations from the American Society for Veterinary Clinical Pathology for sample size >120 [17]. ^2^ ECq reference limits calculated with the non-parametric percentile method; 95% CIs calculated with the bootstrap method based on sample size < 120 [17].

## Data Availability

The data presented in this study are available on request from the corresponding author. The data are not publicly available due to pending publications.

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
