# Peer review of "Evaluation of a Porcine Endogenous Reference Gene (Internal Sample Control) in a Porcine Reproductive and Respiratory Syndrome Virus RT-qPCR"

_vetsci, 2023, doi:10.3390/vetsci10060381_

Round 1
Reviewer 1 Report
Berenice Munguía-Ramírez et al. described the use of a porcine endogenous reference gene in a PRRSV RT-qPCR. They found that the ISC evaluated herein can be used to accurately monitor sample quality in swine specimens tested for PRRSV. The results support these conclusions. Overall, this is an interesting finding. I have some suggestions for improving this manuscript.
1. The author indicated “a porcine specific RNA ISC in a commercial PRRSV RT-qPCR was evaluated for its expression in swine serum, oral fluid, and fecal specimens”. What is the name of this RNA? What is the primer sequence of this RNA? The author must provide more and clearer information.
2. The introduction section is too short. More details about ISCs should be provided.
3. The author conduct the experiment in a commercial PRRSV RT-qPCR. Is it commercial PRRSV RT-qPCR kit? I think the authors should compare the differences between the different kits, especially for study 3.
4. The author should carefully examine English grammar and typography.
5. The title is too complicated and difficult to understand. It should be simple and clear.
The author should carefully examine English grammar
Author Response
- The author indicated “a porcine specific RNA ISC in a commercial PRRSV RT-qPCR was evaluated for its expression in swine serum, oral fluid, and fecal specimens”. What is the name of this RNA? What is the primer sequence of this RNA? The author must provide more and clearer information.
A: The specific gene targeted by the ISC in the commercial PRRSV RT-qPCR kit is unknown as it is proprietary information. However, the kit insert describes the ISC as an “endogenous host RNA”.
Lines 66 – 71 now read: “Herein, a porcine-specific ISC RNA included in a commercial porcine reproductive and respiratory syndrome virus (PRRSV) RT-qPCR (IDEXX Laboratories, Inc.) was evaluated for its expression in swine serum, oral fluid, and fecal specimens. The gene targeted by this ISC is proprietary information, but the product insert describes the target as an “endogenous host RNA” that is amplified simultaneously with PRRSV.”
- The introduction section is too short. More details about ISC should be provided.
A: The introduction has been extended with information about the use of ISC in veterinary diagnostics. However, the authors believe that this information should be concise, not extensive.
- The author conducts the experiment in a commercial PRRSV RT-qPCR. Is it commercial PRRSV RT-qPCR kit? I think the authors should compare the differences between the different kits, especially for study 3.
A: Correct, it is a commercial PRRSV RT-qPCR assay (RealPCR PRRSV Type 1 and Type 2 Multiplex RNA Mix; IDEXX Laboratories, Inc.).
Comparison of different ISCs from different commercial kits would be an important addition to the paper. However, among the commercial PCR kits for swine viruses available in the market at present, the “RealPCRTM” assays (IDEXX) are the only ones including primers and probes for the detection of an ISC.
- The author should carefully examine English grammar and typography.
The authors would appreciate objective observations in the manuscript as English grammar has been previously examined.
- The title is too complicated and difficult to understand. It should be simple and clear.
A: The authors believe that the title expresses concisely the content of the manuscript.
Title now reads: “Evaluation of a porcine endogenous reference gene (internal sample control) in a porcine reproductive and respiratory syndrome virus RT-qPCR”.
Reviewer 2 Report
The manuscript by Berenice et al. entitled “Use of a porcine endogenous reference gene (internal sample control) in a porcine reproductive and respiratory syndrome virus (PRRSV) RT-qPCR” evaluated that the performance of a porcine-specific ISC in a commercial PRRSV RT-qPCR. By ISC porcine specificity, ISC over time in individual pigs, and ISC in field samples three experimental studies. The authors conclude that the ISC response was consistent to the point that failure to detect is sufficient justification for re-testing and/or resampling. However, there are still many minor problems in the manuscript, which need further revision and improvement. The specific amendments are as follows:
1、 It is best not to use English abbreviations in the title, like “PRRSV”.
2、 In the introduction, the description of porcine endogenous reference gene and qPCR detection technology is too simplistic, and this section needs to be expanded.
3、 Please follow the rule that after the first appearance of proper noun, the following proper noun should be abbreviated. For example, ISC has been abbreviated in line 29, and appears again in line 52. The author should carefully revise according to the entire manuscript.
4、 What does “Set 1 and 2” mean in line 33.
5、 It's best to replace “CQ” with “CT” in line 46.
6、 Incorrect format in line 63.
7、 Proper noun that appear only once in an article may not be abbreviated, such as (ASVCP) in line 87 and (AAALAC) in line 111.
8、 Some terms in the article are incorrectly expressed. For example, “TCID50 per mL” should be changed “TCID50/mL”, in line 119.
9、 The clarity of Figures 1A and 1B needs to be improved.
10、Table 1 needs to be redone.
11、The discussion needs to be more in-depth.
12、“To provide guidelines for the interpretation of ISC results, reference limits were calculated for serum, oral fluid, feces, and fecal swabs.”, which is best placed at the beginning of the conclusion.
13、The explanation of the results in the manuscript is insufficient and needs to be added.
14、The DOI in the references needs to be unified, in lines 367, 397, 405, and 432.

English language can be better.
Author Response
Reviewer 2:
The manuscript by Berenice et al. entitled “Use of a porcine endogenous reference gene (internal sample control) in a porcine reproductive and respiratory syndrome virus (PRRSV) RT-qPCR” evaluated that the performance of a porcine-specific ISC in a commercial PRRSV RT-qPCR. By ISC porcine specificity, ISC over time in individual pigs, and ISC in field samples three experimental studies. The authors conclude that the ISC response was consistent to the point that failure to detect is sufficient justification for re-testing and/or resampling. However, there are still many minor problems in the manuscript, which need further revision and improvement. The specific amendments are as follows:
- It is best not to use English abbreviations in the title, like “PRRSV”.
A: Title now reads: “Evaluation of a porcine endogenous reference gene (internal sample control) in a porcine reproductive and respiratory syndrome virus RT-qPCR”.
- In the introduction, the description of porcine endogenous reference gene and qPCR detection technology is too simplistic, and this section needs to be expanded.
A: The authors added more information on ISCs, their use in veterinary diagnostics, and the ISC used in the study. Nonetheless, the authors believe that the information provided in the introduction should remain concise rather than extensive.
- Please follow the rule that after the first appearance of a proper noun, the following proper noun should be abbreviated. For example, ISC has been abbreviated in line 29, and appears again in line 52. The author should carefully revise according to the entire manuscript.
A: The abbreviation in line 29 corresponds to the Abstract, whereas the one in line 52 (now line 50) corresponds to the introduction of the manuscript. Abbreviations are defined in both sections as the Abstract Is independent of the main body of text and must stand alone without reference to the text.
- What does “Set 1 and 2” mean in line 33.
A: We used two sets of samples for Study 2, i.e., Set 1 and Set 2. Set 1 consisted of oral fluid, serum, and fecal samples from pigs vaccinated with a PRRSV modified-live vaccine and followed from -7 to 42 days post vaccination. Set 2 consisted of serum samples from pigs inoculated with PRRSV isolate ATCC VR-2332 and followed from -6 to 182 days post inoculation. We deleted “Set 1 and 2” from the abstract (lines 28 – 41) and addressed the information in lines 113 – 132.
Lines 32 – 34 now read: “In Study 2, the constancy of ISC detection over time (≥ 42 days) was assessed in oral fluid (n = 130), serum (n = 215), and feces (n = 132) collected from individual pigs of known PRRSV status”.
- It's best to replace “CQ” with “CT” in line 46.
A: We appreciate the feedback. However, according to the Minimum Information for Publication of Quantitative Real-Time PCR Experiments (MIQE) guidelines (pp. 361, 336) and Dr. Bustin’s article on the MIQE guidelines (https://doi.org/10.1373/clinchem.2008.112797), the terms “Ct”, “Cp”, or “TOP” referring to the PCR outcomes are based on names provided by the manufacturers in order to differentiate their real-time instruments - they are not driven by scientific considerations. Therefore, the MIQE guidelines (also see http://www.rdml.org) recommend the use of “quantification cycle (Cq)”.
- Incorrect format in line 63.
A: Subheading format in line 64 (previously, line 63) changed to “MDPI_2.2_heading2”.
- Proper nouns that appear only once in an article may not be abbreviated, such as (ASVCP) in line 87 and (AAALAC) in line 111.
A: Acronyms “ASVCP” and “AAALAC” were deleted.
- 8. Some terms in the article are incorrectly expressed. For example, “TCID50 per mL” should be changed “TCID50/mL”, in line 119.
A: Lines 128 – 132 (previously, line 119) now read: “As fully described elsewhere [17], pigs were intramuscularly inoculated with 1 mL (1 x 104 TCID50/mL)”.
- The clarity of Figures 1A and 1B needs to be improved.
A: The authors would like to know the specific points that need to be improved in the figure.
- Table 1 needs to be redone.
A: The authors would appreciate objective observations of the table in order to improve it.
- The discussion needs to be more in-depth.
A: We appreciate the observation, but the authors agreed that the discussion provided a concise summary of the problem addressed, and that supplemental information would be redundant.
- “To provide guidelines for the interpretation of ISC results, reference limits were calculated for serum, oral fluid, feces, and fecal swabs.”, which is best placed at the beginning of the conclusion.
A: The authors relocated the sentence to the beginning of the conclusion:
Lines 326 – 329 now read: “This study evaluated the detection of a porcine-specific ISC in a commercial PRRSV RT-qPCR and provided guidelines for the interpretation of ISC results by calculating reference limits for serum, oral fluid, feces, and fecal swabs collected from pigs under experimental and field conditions.”
- The explanation of the results in the manuscript is insufficient and needs to be added.
A: The authors would appreciate more specific observations in the results section in order to consider adding more information.
- The DOI in the references needs to be unified, in lines 367, 397, 405, and 432.
A:
Note: The manuscript has been modified and the line numbers have changed.
- Line 401 ( Reference #14, Lee et al., 2002). DOI modified.
- Line 413 (#19, Molina-Barrios, 2008). DOI added.
- Line 420 (#22, Martínez-Lobo et al., 2013). DOI added.
- Line 445 (#29, Brattelid et al., 2010). DOI added.

Reviewer 3 Report
In this study, Munguia-Ramirez et al. evaluated the specificity and variability of expression under different conditions (time, matrix, infection...) of a commercial ISC RNA for the RTqPCR diagnosis of PRRSV. This work underlines the importance of using an appropriate CSI according to the pathogen and matrix tested, and, among other issues, it appears crucial to retest negative ISC results. This work contributes to the development of guidelines for viral genome detection in general, and PRRSV in particular.
-lane 226: I guess “oral” is missing after the 2nd percentage.
-lane 237: “PRRSV higher in serum and oral fluids compared to feces”; what are the expectations from the literature?
-lane 253 : 42 days instead 49 days
-Table 1: lanes numbers appear in the tab
Author Response
Reviewer 3:
- Line 226: I guess “oral” is missing after the 2nd percentage.
A: Lines 236 – 237 (previously line 126) now read: “Overall, 158/282 (56.0%) sera, 146/280 (52.1%) oral fluids, and 27/282 (9.6%) of fecal samples were positive for PRRSV RNA.”
- Line 237: “PRRSV higher in serum and oral fluids compared to feces”; what are the expectations from the literature?
It is recognized that PRRSV RNA in fecal samples can be found intermittently, as opposed to serum and oral fluid samples, i.e., lower levels of PRRSV RNA in feces. The manuscript briefly compares our findings with previous literature in lines 237 – 239.
- Line 253: 42 days instead 49 days
A: Pigs were followed from -7 to 42 days post-vaccination (PRRSV modified-lived vaccine). The 49 days are considering the 7 days prior vaccination.
- Table 1: lanes numbers appear in the tab
A: Could the reviewer clarify this point?
Round 2
Reviewer 1 Report
-
The author has revised the manuscript according to my request.
Author Response
No further coments added. The authors thank the reviewer for the observations.
Reviewer 2 Report
1、Figures 1A and 1B are relatively blurry and need to be clearer.
2、Why is the row number in Table 1 on the left instead of the right?
English language can be better.
Author Response
Reviewer 2:
- Figures 1A and 1B are relatively blurry and need to be clearer.
A: The print resolution has been changed from 800 dpi to 1200 dpi. The new image has been inserted in the manuscript and attached as a single zip archive in the revision submission.
- Why is the row number in Table 1 on the left instead of the right?
A: Apologies, but the authors don’t see any row numbers on the left side in Table 1.
English language can be better.
A: This manuscript has been revised and edited by a native English speaker, i.e., Dr. Jeffrey Zimmerman, who has been an editor of several editions of “Diseases of Swine”, a definitive reference book on swine health and disease. After completion, the manuscript went through several revisions by the rest of the authors. Thus, the authors are certain that the grammar has been verified and ask for objective observations on the paragraphs/sentences that need to be improved.
